# Fast Bacterial Succession Associated with the Decomposition of *Larix gmelinii* Litter in Wudalianchi Volcano

**DOI:** 10.3390/microorganisms12050948

**Published:** 2024-05-07

**Authors:** Lihong Xie, Jiahui Cheng, Hongjie Cao, Fan Yang, Mingyue Jiang, Maihe Li, Qingyang Huang

**Affiliations:** 1Institute of Natural Resources and Ecology, Heilongjiang Academy of Sciences, Harbin 150040, China; xielihong903@163.com (L.X.); chengjiahui407@163.com (J.C.); hjcao781228@163.com (H.C.); yangfan81039@163.com (F.Y.); 13804585147@163.com (M.J.); 2Forest Dynamics, Swiss Federal Institute for Forest, Snow and Landscape Research, CH-8903 Birmensdorf, Switzerland; maihe.li@wsl.ch; 3Key Laboratory of Geographical Processes and Ecological Security in Changbai Mountains, Ministry of Education, School of Geographical Sciences, Northeast Normal University, Changchun 130024, China; 4School of Life Science, Hebei University, Baoding 071002, China

**Keywords:** *Larix gmelinii*, bacteria, litter decomposition, WDLC volcano

## Abstract

In order to understand the role of microorganisms in litter decomposition and the nutrient cycle in volcanic forest ecosystems, the dominant forest species *Larix gmelinii* in the volcanic lava plateau of the Wudalianchi volcano was considered as the research object. We analyzed the response of bacterial community structure and diversity to litter decomposition for 1 year, with an in situ decomposition experimental design using litter bags and Illumina MiSeq high-throughput sequencing. The results showed that after 365 days, the litter quality residual rate of *Larix gmelinii* was 77.57%, and the litter N, P, C:N, C:P, and N:P showed significant differences during the decomposition period (*p* < 0.05). The phyla Cyanobacteria and the genus *unclassified_o_Chloroplast* were the most dominant groups in early decomposition (January and April). The phyla Proteobacteria, Actinobacteriota, and Acidobacteriota and the genera *Massilia*, *Pseudomonas*, and *Sphingomona* were higher in July and October. The microbial communities showed extremely significant differences during the decomposition period (*p* < 0.05), with PCoa, RDA, and litter QRR, C:P, and N as the main factors driving litter bacteria succession. Microbial functional prediction analysis showed that Chloroplasts were the major functional group in January and April. Achemoheterotrophy and aerobic chemoheterotrophy showed a significant decrease as litter decomposition progressed.

## 1. Introduction

The process of litter decomposition and soil nutrient release is the key to material circulation and energy transformation in forest ecosystems [1]. During this process, organic materials and nutrient elements are transported to the soil to maintain soil fertility and promote plant growth and development [2,3]. Litter decomposition is a highly dynamic process [4], and a large number of studies have been conducted on ecosystems, including high-frigid forests, alpine meadows, deserts, wetlands, etc. [4,5,6,7], which is important for understanding litter decomposition and controlling factors in different types of ecosystems.

As an important litter decomposer in forest ecosystems, microorganisms facilitate the decomposition of nearly 90% of organic fractions. In boreal and temperate forest soils, the main decomposers are fungi and bacteria, whose community structure and development affect the decay mode of litters [8]. At present, a limited number of studies have been carried out on the microorganism community structure and diversity of litters. Romaní et al. [9] assumed that fungi are very important for the process of litter degradation, whereas bacteria only colonize the soil and benefit from the complex macromolecules produced by soil fungal exoenzymes [10]. Purahong et al. [11] used a molecular biology method to carry out research and observed that the role played by bacteria in the process of litter degradation is underestimated; bacteria also have a significant role in litter decomposition and humus transformation. Different bacteria and fungi co-exist and may provide electrons or necessary micronutrients during the process of litter decomposition to promote the production of fungal degradation products [12]. Therefore, bacteria play an indispensable role in litter degradation.

Bacteria have a significant role in litter decomposition, C:N mineralization, humus formation, nutrient conversion, and other material cycles [13]. Litter decomposition is influenced by multiple factors, including the substrate quality of litters, the local climate, the microorganism community structure, the soil texture, etc. [14,15,16,17]. The bacterial community in litter showed significant seasonal dynamics in forest ecosystems in response to soil moisture and temperature [18]. Litter decomposition is a highly dynamic process [4], and the composition of microorganisms involved in litter decomposition varies depending on the stage of decomposition, which can have implications for the process of decomposition [4,19].

The Wudalianchi (WDLC) volcano erupted on a large scale from 1719 to 1721 and formed a lava plateau with an area of 65 km^2^, with a thin litter layer and poor soil [20]. The nutrients and water conditions required for plant growth are scarce [21]. After the eruption, the renewal of vegetation succession and the biogeochemical cycle changed and formed a volcanic forest ecosystem with special characteristics, landforms, soils, and hydrology [22,23,24]. As an outstanding example of a volcanic ecosystem, the WDLC volcano has attracted the attention of an increasing number of scholars. This study investigates the successional changes in bacterial communities during the decomposition of *Larix gmelinii* litters in the WDLC volcanic lava plateau. We hypothesized that litter microbial community structures differ in different decomposition stages. Therefore, we assessed the following hypotheses: (1) The variations in litter quality and nutrient affect the structure and function of microbial communities during litter decomposition. (2) After the litter falls to the ground and decompose, phyllosphere-related organisms prevail initially and then are replaced by those coming from soil. The results obtained in this study will provide a theoretical basis for the study of the plant–litter–soil nutrient cycle in volcanic forest litter systems.

## 2. Materials and Methods

### 2.1. Overview of the Study Area

The study area is located at WDLC National Park, Heihe, Heilongjiang Province. WDLC has a temperate continental monsoon climate with an annual average temperature of −0.5 °C, an annual average precipitation of 476.33 mm, an annual average relative humidity of 69.2%, and a frost-free period of 121 days [8]. The lava plateau has volcanic rocky soil (Figure 1). Its belongs to the temperate and boreal mixed (needle and broad leaves) forest land with Ass. *Larix gmelinii* and *Betula platyphylla* mixed forest as the main vegetation type, the important species are *Larix gmelinii*, *Betula platyphylla*, *Populus davidiana*, *Populus koreana*, etc. [20].

### 2.2. Experimental Design and Sample Analysis

The litter of *Larix gmelinii*, which is the dominant tree at the WDLC volcanic lava plateau, was collected in late September 2021. The litters were blended evenly at room temperature and dried to a constant weight. And the 10.0 g samples were weighed and packed into a litter bag of 100 mesh, 35 cm × 25 cm in size. Three unconnected and flat sites were selected as 3 replicates for the decomposition experiment in the study area. Each decomposition experimental site covered an area of 50 m^2^. In mid-October 2021 (by the first snow), we removed litter and debris from the soil surface and laid litter bags on the surface of the sample plot flat, with each litter bag spaced at least 5 cm apart. The bags were kept as close to the natural state as possible and fixed with plastic ground nails.

### 2.3. Sample Collection

In the experiment, sampling was conducted four times in January 2022, April 2022, July 2022, and October 2022 (Figure 2). The first sampling to the last sampling lasted for one year. During sampling, one bag was used for each treatment at each decomposition site. Three bags were collected at each of the sites, heated at 105 °C to sterilize enzymes, and then dried at 80 °C to constant weight; the mass residual rate of litters was calculated. The dried samples were ground and sieved, and the carbon, nitrogen, and phosphorus contents of the dried samples and samples (T_0_) before decomposition were measured. Meanwhile, three replicate samples were collected and stored in liquid nitrogen to measure bacteria in litters. Litter DNA extraction and high-throughput sequencing of bacteria were entrusted to Shanghai Major Biomedical Technology Co., Ltd. (Shanghai, China).

### 2.4. Measurement

The litter total nitrogen (N) and organic carbon (C) were determined by an elemental analyzer (EA3000, Euro Vector, Foggia, Italy). The total phosphorus (P) mass fraction of the soil was determined using molybdenum antimony colorimetry. Total DNA of litter was extracted using a MoBio PowerSoil kit (MoBio, Carlsbad, CA, USA), and 16sRNA amplification primers for bacteria were used using universal primers 515F (5′-GTFYCAGCMGCCG CGGTAA-3′) and 806R (5′-GGACTACNVGGGTWTCTAAT-3′). Species classification was compared and identified using the bacterial database of silver 128/16s bacteria. High-throughput sequencing was commissioned to Shanghai Major Biomedical Technology Co., Ltd. (Shanghai, China). Bioinformatics analysis was completed using the microbial diversity V4.0 cloud platform.

### 2.5. Statistical Analysis

SPSS 19.0 was used to analyze the quality residual rate, physicochemical properties, microbial diversity, and community of litter during different decomposition stages in one-way ANOVA (*p* < 0.05). Least significant difference (LSD) test was also used for multiple comparisons. R (Team, 2019) was used for statistical computations. Based on the vegan package, we used principal co-ordinate analysis (PCoA) analysis to ascertain the bacteria community structure, and redundancy analysis (RDA) to visualize the influence of litter variables on bacterial community composition. Data are reported as the mean ± standard error.

## 3. Results

### 3.1. Shifts in Nutrient Elements with the Decomposition of Plant Litter

After 365 days of decomposition (Table 1), fast decomposition of *Larix gmelinii* was observed in plant litter (22.43% mass lass). The quality residual rate decreased as the decomposition progressed (*p* < 0.05). Plant litter total phosphorus (P) and C:N showed similarly varied trends to the litter quality residual rate. Litter N, C:P, and N:P showed a significant increase during the whole decomposition period. ANOVA showed that sampling dates had significant effects on plant litter N, P, C:P, C:N, and N:P (*p* < 0.05).

### 3.2. Shifts in Bacterial Community with the Decomposition of Plant Litter

At the phylum level, there were seven bacterial phyla with relative abundances above 1% (Figure 3). The Proteobacteria (11.58–73.67%) and Cyanobacteria (1.36–87.03%) were the most abundant groups; other abundant phyla were Actinobacteriota (0.30–12.81%), Acidobacteriota (0.33–11.32%), Bacteroidota (0.41–5.54%), and Myxococcota (0.18–1.18%). Proteobacteria (67.01–73.67%) was the most abundant group in July and October, higher than in January and April. Cyanobacteria (76.14–87.03%) was the most abundant group in January and April, and showed a significant decline as litter decomposition progressed (*p* < 0.05). More, the relative abundance of Actinobacteriota, Acidobacteriota, and Bacteroidota showed a significant increase as litter decomposition progressed (*p* < 0.05).

We observed significant differences in the communities’ composition at the genus level: *unclassified_o_Chloroplast* (76.13–87.03%) was the dominant group during the initial litter decomposition stage (January and April), and it decreased as litter decomposition progressed (*p* < 0.05; Figure 4). However, *Massilia*, *Pseudomonas*, *Sphingomonas*, *Luteibacter*, *Terriglobus*, *Mucilaginibacter*, and *Robbsia* were higher in July and October than in January and April (*p* < 0.05).

### 3.3. Microbial Diversity Changes during Litter Decomposition

The sequencing and subsequent cleaning and filtering of the reads produced 674,034 useful sequences (56,169 sequences per sample) across all the samples, with an average length of 375 bp. In total, the sequences belonging to 1127 different OTUs were assigned to bacteria, and the bacterial diversity index of the Sobs ranged from 195 to 468 as the decomposition progressed (*p* < 0.05, Table 2). Also, the bacterial diversity indices of Shannon, Ace, and Pd increased as the decomposition progressed (*p* < 0.05).

### 3.4. β-Diversity

Clear differences in the bacterial communities in the plant litter were observed at different decomposition stages (Figure 5). The PERMANOVA analysis (main test) also showed that the bacterial communities were significantly different between the different sampling dates or seasons (R^2^ = 0.84, *p* = 0.001). The close clustering of plant litter samples for the January and April sampling dates indicated similar bacterial community compositions. The samples from July and October were distinctly separated from the other sampling dates. As the decomposition progressed, the average Bray-Curtis dissimilarity of bacterial communities increased from 27.67% (January, 92 days) to 66.26% (October, 365 days).

### 3.5. RDA Analysis

At the OUT level, RDA analysis was conducted on bacterial communities at all decomposition stages (Figure 6). Using the inflation factor (VIF) test to screen for reduced collinearity, QRR (VIF = 4.362), C (VIF = 1.161), N (VIF = 2.195), and C:P (VIF = 2.806) were obtained. RDA1 and RDA2 described 87.92% and 0.42% of the discrepancies in litter bacterial community structure. The driving factors that significantly affected the bacterial community structure were QRR (R^2^ = 0.976, *p* = 0.001), C:P (R^2^ = 0.606, *p* = 0.021), and N (R^2^ = 0.5776, *p* = 0.019). The samples were clearly separated between sampling dates, indicating that the bacterial decompositions were distinct at different decomposition stages. The results from VIF and RDA suggested that litter QRR, C:P, and N were the most relevant variables that affect the litter bacterial community structure.

The litter bacterial community structure closely correlated with the litter physicochemical properties (Figure 7a). Protcobacteric, Actinobacteriota, Acidobacteriota, and Bacteroidota were significantly positively correlated with litter N, C:P, and N:P, and negatively correlated to QRR, P, and C:N (*p* < 0.01), Cyanobacteria was negatively correlated with litter N, C:P, and N:P, and positively correlated with QRR, P, and C:N (*p* < 0.01). Myxococcota was correlated with litter N (*p* < 0.05).

At the genus levels, unclassified_o_Chloroplast was significantly positively correlated with QRR, P and C:N (*p* < 0.05), and significantly negatively correlated with N, C:P, and N:P (*p* < 0.05). Some advantageous genera, such as *Massilia*, *Pseudomonas*, *Sphingomonas*, *Luteibacter*, *Terriglobus*, *Mucilaginibacter*, *Amantichitinum*, and *Robbsia*, were significantly negatively correlated with QRR, P, and C:N (*p* < 0.05), and significantly positively correlated with N, C:P, and N:P (*p* < 0.05, Figure 7b).

### 3.6. Bacterial Function Prediction in Litter

FAPROTAX was used to predict and annotate the function of the litter bacterial community. There were eight functional groups with relative abundances greater than 1% in the bacterial communities (Table 3). All bacterial functional groups were significantly different (*p* < 0.05). Chloroplasts were the major functional group, and had the greatest value in January. Secondly, achemoheterotrophy and aerobic chemoheterotrophy were also major functional groups, and had the greatest value in October. More, ureolysis, animal parasites or symbionts, human pathogens all, and human pathogens pneumonia were significantly higher in July and October than in January and April, while intracellular parasites were significantly higher in January and April than in July and October.

## 4. Discussion

### 4.1. Variation in Nutrient Elements before and after Litter Decomposition

Litter decomposition rates closely correlate with their corresponding chemical properties [25]. After 365 days of decomposition, the N content in *Larix gmelinii* was significantly higher than that before decomposition. Previous studies showed that during early decomposition, microorganisms use litter as the nutrient provider. When the N in the litter cannot meet the decomposition needs of microorganisms, a certain amount of N needs to be absorbed from the external environment (soil or precipitation) to form microbial biomass or exoenzymes, resulting in a net increase in the N content in litters [26]. The higher N content in plant litter caused a significant decrease in C:N ratio in plant litter [27]. A similar result was found in our study: compared with the initial C:N ratio in plant litter, it significantly decreased from 29.25 to 20.33.

Given the differences in the effectiveness of N and P in different ecosystems, N:P can also be used as the standard of litter decomposition limit [28]. In this study, the N:P of *Larix gmelinii* was less than 14, which indicates that litter decomposition was largely limited by N content. This finding is consistent with the results of our research on the dominant plant *Populus davidiana* at the WDLC volcano [29]. The lava plateau at WDLC is faced with problems, including sparse vegetation, poor soil, and nitrogen deficiency, which further verified that the growth of volcanic plants and litter decomposition are limited by N deficiency.

### 4.2. Differences in Bacterial Community Structure and Diversity in Litter

In the process of litter decomposition, Cyanobacteria (67.01–87.03%) and *unclassified_o_Chloroplast* (76.13–87.03%) were the dominant groups in January and April, and showed a significant decline as litter decomposition progressed. Among diverse species, members of Cyanobacteria are frequently detected in the phyllosphere, including both leaf endophytes and epiphytes as well as those saprobic bacteria commonly isolated from plant debris [11,30]. It suggests that phyllosphere microorganisms were the dominant groups in early decomposition, but as the decomposition progressed, soil bacteria entered the litter layer through raindrop splashing, wind action, and the carrying of fungal mycelium [31]. In this study, Proteobacteria, Actinobacteriota, and Acidobacteriota were the dominant nitrogen-fixing microorganisms in July and October, which is consistent with the results of soil bacterial communities in long-term time series [24], and exhibit environmental filtration [32]. Actinobacteria is a nutrient-poor bacterial community, which prefers nutrient-poor environments and has a higher concentration in extreme habitats [33].

The variation in microbial community structure is evident at the genus level. In the early stage of litter decomposition, *Massilia*, *Pseudomonas*, *Sphingomonas*, and *Methylobacterium* were among the dominating genera; these members were detected in litters during decomposition in other forest ecosystems [18,34]. The bacterial community structure at the genus level was surprisingly similar to that found at the corresponding stages of decomposition in *Quercus wutaishanica* litter [4]. *Sphingomonas* was dominant in the early decay litters (9.26–16.83%). It can use all sorts of organic matters, including soluble and relatively complex types, to produce proteolytic or cellulolytic enzymes [35,36]. Some members of *Methylobacterium* were frequently identified as dominant taxa in the phyllosphere of various plants, where the bacteria of soil invade and participate in litter decomposition; this pattern was consistent with the bacterial succession during the decomposition of litters [37]. Overall, these research results suggest that the composition of microbial communities responds sensitively to the decomposition process and plays a positive role in the forest ecosystem.

### 4.3. Relationship between Nutrient Elements and Bacterial Community Structure in Litters

Litter decomposition is the core process in nutrient cycling [38]. C and N in litter drive bacterial succession during the decomposition process [39,40]. Our research showed that Proteobacteria and Bacteroidetes were closely related to the C:N in litter decomposition, which mainly use easily decomposable carbon as the carbon source. The abundance of diazotrophs [5] such as Proteobacteria and Bacteroidetes showed a significant increase as decomposition proceeded in the litter, which directly led to a continuous increase in litter nitrogen content and explained the decrease in the litter C:N ratio [41,42].

The existence of N-fixing microbes results in the accumulation of plant litter N [43]. In our study, *Massilia*, *Pseudomonas*, and *Sphingomonas* were found to be significantly related with litter N (*p* < 0.05); these can use some complex substrates or fix nitrogen, and varied during the decomposition process [44], which may have accounted for the increase in the total N content in the litter. In addition, the growth of most bacteria is connected with the need for P for ribosomal RNA synthesis, which promotes the bacterial community to enter the litters [45]. However, compared with N, the relationship between P and microbial structure and function is less known [46].

### 4.4. Potential Functional Groups of Soil Microbes in Litter Decomposition

Current evidence indicates that changes within bacterial communities play a critical role in carbon and nitrogen cycling [47]. In our study, chemoheterotrophy and aerobic chemoheterotrophy are the dominant bacterial functional groups, which are mainly involved in leaf litter’s carbon cycle; this result is consistent with Zhang et al. [48]. There was no significant difference in the C content of litter at different decomposition stages of *Larix gmelinii*, but chemoheterotrophy and aerobic chemoheterotrophy significantly increased with the progress of decomposition, which indicates that many bacteria cannot fix carbon, some show that some bacteria groups obtain carbon and energy by oxidizing organic matter for their growth promoting the assimilation and utilization of leaf litter carbonby bacteria [47,49]. The functional groups involved in the nitrogen cycling process are urea degradation, with *Mesorhizobium*, *Masilia* and *Roseomonas* are dominant, which is the same results in Mount Huangshan [50,51].

## 5. Conclusions

We conducted a litter bag experiment using the Illumina Miseq sequencing method to analyze bacterial community dynamics during the litter decomposition process. During the litter decomposition of *Larix gmelinii*, litter quality significantly decreased and microbial succession occurred rapidly. The phyllosphere bacteria were the dominant group in early decomposition; after 273 days of decomposition, the phyllosphere communities were replaced by new communities with distinct compositions, and exhibited environmental filtration. Bacterial communities may be controlled by both the litter quality and environmental factors (C:P and N), but further research is needed to understand which organisms are responsible for particular processes during decomposition, which will contribute to a deeper understanding of the microbial mechanisms of litter decomposition and litter element cycling.

## Figures and Tables

**Figure 1 microorganisms-12-00948-f001:**
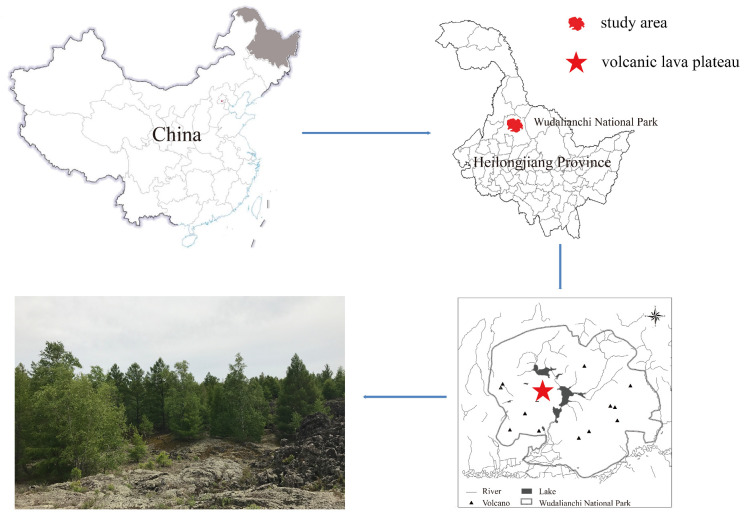
Overview of the WDLC study area. The asterisk indicates the study site in the WDLC National Park, Heilongjiang Province, China.

**Figure 2 microorganisms-12-00948-f002:**
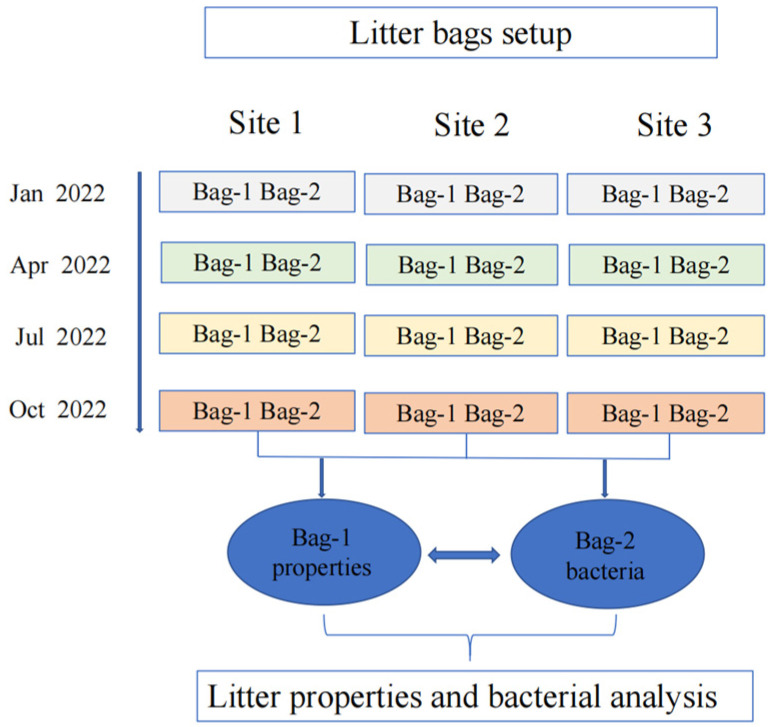
The setup of the litter decomposition experiment.

**Figure 3 microorganisms-12-00948-f003:**
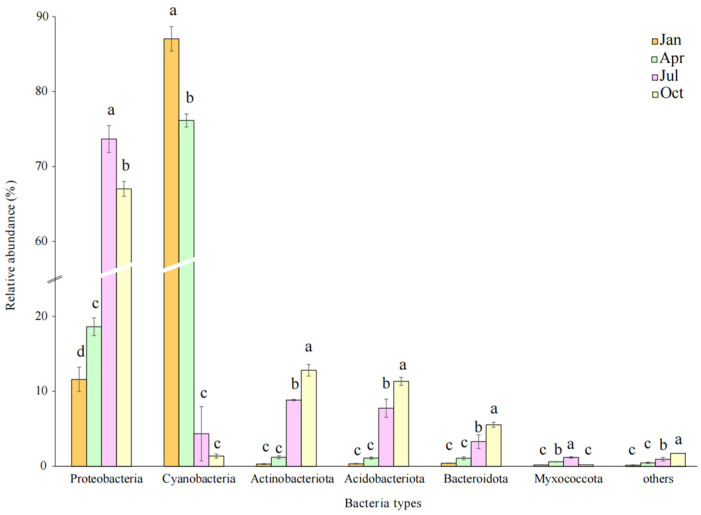
Relative abundance at the phylum level during the decomposition process. If the letter is the same, the divergence is not significant; on the contrary, divergence is significant at the 0.05 level.

**Figure 4 microorganisms-12-00948-f004:**
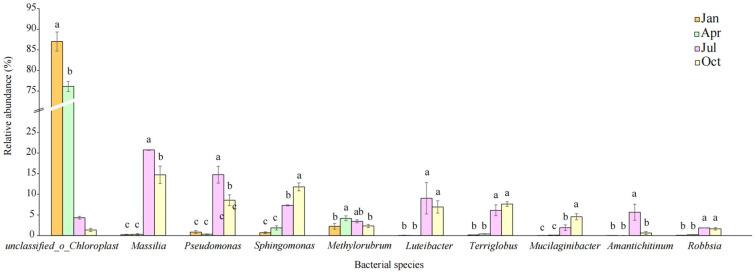
Relative abundance at the genus level during the decomposition process. If the letter is the same, the divergence is not significant; on the contrary, divergence is significant at the 0.05 level.

**Figure 5 microorganisms-12-00948-f005:**
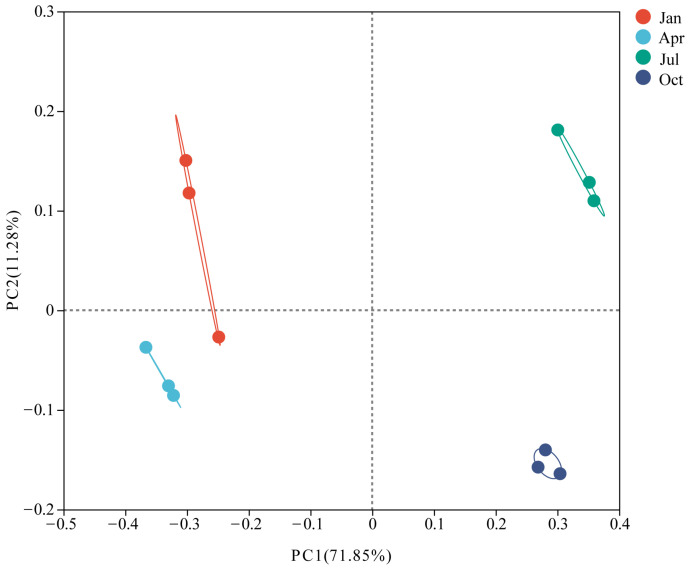
Analysis of PCoA based on the OTU data between different sampling dates.

**Figure 6 microorganisms-12-00948-f006:**
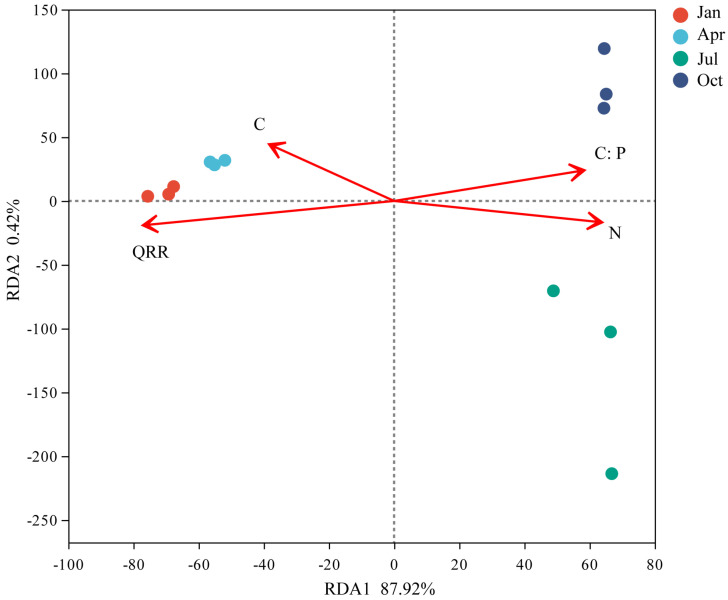
Redundancy analysis (RDA) of the variation in litter bacterial community structure as explained by litter properties.

**Figure 7 microorganisms-12-00948-f007:**
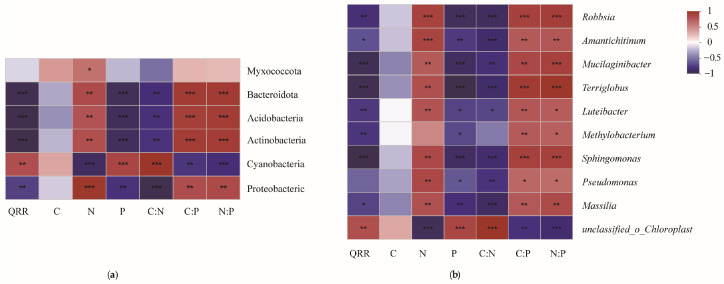
Spearman’s rank correlation coefficients describing the relationships between litter nutrients and microbial community compositions at the phylum level (**a**) and the genus levels (**b**). (*** *p* < 0.001), (** *p* < 0.01), and (* *p* < 0.05).

**Table 1 microorganisms-12-00948-t001:** Plant litter quality residual rate and nutrient elements across different decomposition stages. The data in the table represent mean ± standard error (*n* = 3). If the letter is the same, the divergence is not significant; on the contrary, divergence is significant at the 0.05 level.

Sampling Date	Quality Residual Rate (%)	C (g kg^−1^)	N (g kg^−1^)	P (g kg^−1^)	C:N	C:P	N:P
Initial content (0 d)	100 ^a^	48.91 ± 0.23 ^a^	1.67 ± 0.04 ^c^	3.01 ± 0.03 ^a^	29.25 ± 0.8 ^a^	16.27 ± 0.24 ^b^	0.56 ± 0.02 ^c^
January (92 d)	89.60 ± 0.11 ^b^	49.63 ± 0.39 ^a^	2.09 ± 0.05 ^b^	3.04 ± 0.05 ^a^	23.79 ± 0.36 ^b^	16.34 ± 0.35 ^b^	0.69 ± 0.02 ^c^
April (182 d)	87.30 ± 0.35 ^c^	49.79 ± 0.28 ^a^	2.37 ± 0.01 ^a^	2.84 ± 0.02 ^ab^	21.02 ± 0.16 ^c^	17.53 ± 0.04 ^b^	0.83 ± 0.01 ^b^
July (273 d)	82.03 ± 0.51 ^d^	49.18 ± 0.45 ^a^	2.45 ± 0.00 ^a^	2.63 ± 0.04 ^bc^	20.06 ± 0.2 ^c^	18.73 ± 0.14 ^ab^	0.93 ± 0.01 ^ab^
October (365 d)	77.57 ± 0.51 ^e^	49.27 ± 0.08 ^a^	2.42 ± 0.01 ^a^	2.37 ± 0.21 ^c^	20.33 ± 0.11 ^c^	21.14 ± 1.85 ^a^	1.04 ± 0.09 ^a^

**Table 2 microorganisms-12-00948-t002:** Litter bacterial diversity indices under different decomposition periods. If the letter is the same, the divergence is not significant; on the contrary, divergence is significant at the 0.05 level.

Sampling Date	Sobs	Shannon Index	ACE Index	Pd
January (92 d)	195 ± 35.34 ^c^	0.79 ± 0.128 ^d^	368.3 ± 50.39 ^c^	19.77 ± 3.63 ^c^
April (182 d)	336 ± 25.12 ^b^	1.409 ± 0.048 ^c^	536.5 ± 83.1 ^b^	34.71 ± 1.93 ^b^
July (273 d)	350.7 ± 52.35 ^b^	3.28 ± 0.212 ^b^	674.2 ± 66.6 ^ab^	31.17 ± 3.15 ^b^
October (365 d)	468.7 ± 22.81 ^a^	3.888 ± 0.053 ^a^	705.6 ± 103.8 ^a^	42 ± 5.73 ^a^

**Table 3 microorganisms-12-00948-t003:** Differences in potential functional groups with average relative abundance > 1%. The data in the table are mean ± standard error (*n* = 3). If the letter is the same, the divergence is not significant; on the contrary, divergence is significant at the 0.05 level.

Functional Groups	January	April	July	October
chloroplasts	90.69 ± 1.04 ^a^	83.04 ± 1.09 ^b^	0.64 ± 0.06 ^c^	1.02 ± 0.29 ^c^
achemoheterotrophy	2.39 ± 0.28 ^c^	4.92 ± 0.3 ^b^	23.93 ± 2.66 ^a^	24.73 ± 2.55 ^a^
aerobic chemoheterotrophy	2.07 ± 0.24 ^c^	3.66 ± 0.58 ^b^	23.00 ± 1.75 ^a^	24.09 ± 2.49 ^a^
ureolysis	0.23 ± 0.04 ^c^	0.87 ± 0.10 ^b^	13.61 ± 0.47 ^a^	11.69 ± 1.39 ^a^
animal parasites or symbionts	0.06 ± 0.01 ^c^	0.18 ± 0.05 ^b^	11.95 ± 1.59 ^a^	10.81 ± 1.19 ^a^
human pathogens all	0.06 ± 0.01 ^c^	0.15 ± 0.02 ^b^	11.93 ± 1.59 ^a^	10.81 ± 1.19 ^a^
human pathogens pneumonia	0.02 ± 0.00 ^b^	0.03 ± 0.00 ^b^	12.59 ± 0.72 ^a^	10.77 ± 1.19 ^a^
intracellular parasites	3.45 ± 0.40 ^a^	3.47 ± 0.47 ^a^	0.38 ± 0.03 ^b^	0.57 ± 0.06 ^b^

## Data Availability

Data available upon request.

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
