# Peer review of "Fast Bacterial Succession Associated with the Decomposition of Larix gmelinii Litter in Wudalianchi Volcano"

_microorganisms, 2024, doi:10.3390/microorganisms12050948_

Round 1

Reviewer 1 Report

Comments and Suggestions for Authors

The research described by the authors may be interesting to some readers.

The manuscript is mostly well written in reports of the used methods and obtained results.

However, there are some drawbacks which in my opinion should be addressed by the authors. Mainly, more details on the types of statistical analysis should be provided, as sometimes it is not clear what tests were performed.

Section 3.1

"After 365 days’ decomposition (Table 1), fast decomposition of Larix gmelinii was observed in plant litter (33.43% mass lass)."

To which column in Table 1 it is related? I cannot notice their mass. The authors presented in the first column "quality residual rate" but it is presented as percent (%). If it is this column, a reduction from 100 to 77.57 is not 33.43%.

What statistical tests were used for testing differences between observations in each of the columns in Table 1? Are the assumptions of statistical tests reasonable?

The same should be described for data presented in other figures and tables. The authors report p<0.05. However, for parametric tests, this quantity is calculated when the assumptions of the statistical test are met. 

In my opinion, some of the charts presented by the authors could be improved to more clearly show the phenomenon to which the authors want to draw attention. I hope I read the authors' intentions correctly. But it is just a suggestion and not a strong requirement. For example Figure 1 on the x-axis there could be time and various phyla represented by lines. That would demonstrate the trends and help in comparison between the phyla.

Probably a similar suggestion to Figure 1.

Reviewer 2 Report

Comments and Suggestions for Authors

Regarding the manuscript ID microorganisms-2998946.

      The authors have analyzed the response of bacterial community structure and diversity to liÄ´er decomposition during 1 years, using litter bags and Illumina MiSeq high-throughput sequencing. It is very interesting for the field of soil microbiology in extreme habitats, such as Volcanos. The results regarding the bacterial community and litter quality residual rate and nutrient elements are original and relevant for the field of Soil Microbiology. I would suggest to better clarify the main hypothesis of this study adding scientific statements.

      The results of this manuscript effectively provide insights into extreme habitats regulating microbial community. This is particularly interesting as the authors combined results from litter bags and Illumina MiSeq high-throughput sequencing. However, the quality of the experimental desing could be improved if the authors provided a figure illustrating the experimental desing and all steps from sampling to lab analysis.

      It adds an interesting dataset from field observation in a extreme habitat (Volcano), which can be considered a short-term field experiment. Additionally, the authors have compiled a database from litter bags and Illumina MiSeq high-throughput sequencing. This is highly pertinent and provides a deeper understanding of how extreme conditions modulate microbial community and organic matter decomposition, but it must be better justified in the discussions section. The experimental design is robust, and there is no need to add further controls. I would suggest incorporating a figure as previously described.     

The conclusions are consistent with the aim of the study regarding the litter bag experiment using Illumina Miseq sequencing method to analyze bacterial community dynamics during the liÄ´er decomposition process. All questions were addressed in the results and further discussed accordingly. Perhaps they need to better discuss how extreme conditions modulate microbial community and organic matter decomposition.

      Tables and figures follow the authors' guidelines provided in the Microorganisms' website. All figures have high resolution (300 dpi) and are easy to understand. Tables provide enough information and are well-presented. However, I reccommend using a figure illustrating the experimental desing and all steps from sampling to lab analysis.

Reviewer 3 Report

Comments and Suggestions for Authors

The manuscript entitled "Fast bacterial succession associated with the decomposition of Larix gmelinii liÄ´er in Wudalianchi Volcano" is in the journal Microorganisms. The manuscript is well written.

"Introduction" chapter - is sufficient to justify the topic.

 "Materials and Methods" chapter does not raise any methodological concerns. The methods used are correct.

"Results" chapter - the results are well described and presented in tables and figures.

"Discussion" chapter provides a good commentary on the results obtained in the light of the scientific literature.

"Conclusion" chapter - the conclusions drawn from the research carried out.

"References" chapter - includes the necessary literature in the field of the presented research.
